# Review of Developed Methods for Measuring Gas Uptake and Diffusivity in Polymers Enriched by Pure Gas under High Pressure

**DOI:** 10.3390/polym16050723

**Published:** 2024-03-06

**Authors:** Jae Kap Jung

**Affiliations:** Hydrogen Energy Group, Korea Research Institute of Standards and Science, Daejeon 34113, Republic of Korea; jkjung@kriss.re.kr; Tel.: +82-42-868-5759

**Keywords:** gas uptake, diffusivity, gravimetric method, volumetric method, manometric method, gas chromatography, validation

## Abstract

Gas emission and diffusion through polymeric materials play crucial roles in ensuring safety and monitoring gas concentrations in technology and industry. Especially, the gas permeation characteristics for O-ring material should be investigated for sealing application in a hydrogen infrastructure. To accommodate the requirements of different environments, we first developed four complementary effective methods for measuring the gas absorption uptake from polymers enriched by pure gas under high pressure and determining the gas diffusivity. The methods included the gravimetric method, the volumetric method, the manometric method, and gas chromatography, which are based on mass, volume, pressure, and volume measurements, respectively. The representative investigated results of the developed methods, such as gas uptake, solubility, and diffusivity are demonstrated. The measuring principles, measuring procedures, measured results, and the characteristics of the methods are compared. Finally, the developed methods can be utilized for testing transport properties, such as the leakage and sealing ability, of rubber and O-ring material under high pressure for hydrogen fueling stations and gas industry.

## 1. Introduction

Gas permeation in materials plays a crucial role in industrial applications [1,2,3,4,5,6,7] in various fields, such as polymer electrolytes, batteries, catalysts, protective coatings, fuel cells, gas membrane separation, gas monitoring sensors, food packing, gas storage vessels, and O-ring seals. Permeation is the penetration of a permeate through a polymer membrane. Gas permeability is a physical and chemical process [8,9,10] that includes the absorption, diffusion, and desorption of gas molecules. Permeation works through diffusion, which is the key process for determining the permeability and solubility of a gas. The diffusion process is modeled by Fick’s law of diffusion [11,12,13]. Diffusion is the process in which permeation is driven by a gas concentration gradient. Thus, the gas molecules move from high concentration regions to low concentration regions.

There are various methods for characterizing gas transport properties, such as solubility, diffusivity, and permeability [14,15,16,17,18,19]. Generally, researchers utilize the differential pressure, gas chromatography, and magnetic suspension balance methods.

A standard method for determining gas permeability is the differential pressure method, using a gas permeation test cell according to ISO 15105-1 [18,20,21,22]. The permeation cell consists of two cells separated by a testing sheet. One cell is the high feed concentration side, which receives the testing gas from the feed and vents it. The second cell is the lower permeate concentration side, which receives the corresponding gas and transports it to a gas pressure detector. This commercial apparatus has been utilized in various applications. However, the equipment, including the vacuum pumps, is rather complex and specific to certain gases. Under steady state conditions, the method generates permeability, diffusion coefficient, and solubility data.

Gas chromatography is also an appropriate method to qualitatively and quantitatively evaluate gases [23,24]. Detectors such as pulsed discharge detectors (PDDs) and thermal conductivity detectors (TCDs) can be used depending on their sensitivity to the single gas or mixed gas to be analyzed. Another detection method for gas transport properties is the volumetric method, typically with differential pressure applied in the cell [25]. However, these methods require large instruments, elaborate and fine control strategies, vacuum control/monitoring, and periodic maintenance.

In previous works, four simple and effective methods for measuring the time-varying gas concentration released from polymer specimens enriched by various gases in the desorption process were developed, and the diffusivity was determined by employing a dedicated diffusion analysis program. The methods are based on the gravimetric measurement [26,27], volumetric measurement [27,28], manometric measurement, and thermal desorption analysis–gas chromatography (TDA–GC) method [24,27] for quantitative evaluation of the released gas. The gravimetric method (GM) uses commercial electronic balances with minute resolution to quantize the amount of gas emitted in environments with well-maintained temperature and humidity. The volumetric method (VM) monitors decreased water levels in graduated cylinders to measure the uptake and diffusivity of the emitted gas. We employed a manometric method (MM) based on pressure measurements in a gas-enriched sample container with a simple pressure logger. The increase in gas pressure caused by the released gas was used to measure the gas concentration and diffusivity with the data pressure logger. Finally, we established an elaborate GC procedure to determine the H_2_ transport parameters for H_2_ emitted from polymeric materials enriched under high-pressure conditions.

The previous research was related the diffusivity and solubility obtained from the experiments for five gases (H_2_, He, N_2_, O_2,_, and Ar) to the kinetic diameter and critical temperature, respectively. These gases could be appropriate candidates for the investigation of gas transport properties, because they were cheap and easily available. In this review article, the representative three types of gases (H_2_, N_2_, and O_2_) in the experiment and analysis were chosen. The measuring principles, measuring procedures, and representative results for the three gases obtained by four methods are described. The performance and characteristics of the GM, VM, MM, and GC method are reviewed. They included the investigations of various gases uptake and diffusivity obtained by methods. The comparisons among the four methods, together with features of these methods, are also contained.

## 2. Sample Preparation and Gas Exposure under High Pressure Conditions

To measure gas uptake and diffusivity in polymeric rubber specimens filled with carbon black, such as ethylene propylene diene monomer (EPDM), nitrile butadiene rubber (NBR), and fluoroelastomer (FKM), used as sealing materials in the O-ring, are employed in this work. Low-density polyethylene (LDPE) was also included as a specimen for experimental investigations. The composition and density of the specimens are listed in previous literatures [24,29]. NBR, EPDM, FKM, and LDPE specimens with cylindrical and spherical shapes are used.

The heat treatment was conducted at 343 K for more than 48 h under the atmosphere according to the CHMC 2 standard [30] to remove the outgassing from the rubber specimen. Then the relative change in the mass of the specimen was measured using an electronic balance, and the measured change was less than 5 wt∙ppm over 24 h. This confirmed that outgassing from the specimen was completely removed.

A 316 stainless steel (SUS) chamber with an outer rectangular shape (length 150 mm × width 100 mm × height 150 mm) and inner cylindrical shape (diameter 50 mm × height 60 mm), as shown in Figure 1, was used for gas exposure under high pressure conditions at 298 K. The pressure chamber was purged at least three times with the corresponding gas below 3 MPa before testing the gas exposure. Then we exposed the specimen to the gas for 24–36 h at the corresponding pressure. Gas exposure for 36 h was regarded as sufficient for reaching equilibrium for gas absorption. The purities of the pure gases in the review work are as follows: H_2_: 99.99%, He: 99.99%, N_2_: 99.99%, O_2_: 99.99%, and Ar: 99.99%.

## 3. Measuring Principle and Procedure for the Four Methods

We describe briefly the measuring principle and procedure of the four methods, such as GM, VM, MM, and GC. In the last section, the diffusion analysis program for obtaining diffusion parameters and algorithm, which is applied commonly to all methods, is demonstrated.

### 3.1. Gravimetric Measurement of the Gas Emitted by Enriched Specimens

After exposure to H_2_ gas for the set time, the gas in the chamber (Figure 1 and Figure 2) was released by opening the needle valve. After decompression, the elapsed time is recorded from the moment (*t* = 0) at which the high pressure of the gas chamber decreased to atmospheric pressure.

As depicted in Figure 2, the concentration of H_2_ gas released from a specimen is measured in real time by electronic balances with a resolution of 10 μg with a GPIB-interfaced PC over appropriate time intervals until mass equilibrium was reached. An electronic balance (Figure 2) is placed in a stable temperature and humidity chamber, with the temperature and humidity controlled and maintained within 298 ± 1.0 K and 10 ± 3%, respectively. The residual mass (*C_R_*) of the specimen versus the elapsed time is calculated as follows [27]:(1)$$C_{R}\left(t\right)\left[wt\cdot ppm\right]=\frac{M\left(t\right)-M_{0}}{M_{0}}\times 10^{6}$$
where M(t) is the mass of the specimen after decompression for an elapsed time t. and $M_{0}$ is the mass of the specimen before H_2_ exposure in the high-pressure chamber. The inevitable time delay (lag) between decompression and the start of the mass measurement is approximately 5 min. Thus, the missing H_2_ content released from *t* = 0 min to *t* = 5 min after decompression is determined by extrapolation in the diffusion analysis program, which is described later.

### 3.2. Volumetric Measurement of the Gas Emitted by Enriched Specimens

A VM using a graduated cylinder is utilized, as shown in Figure 3. After gas exposure in the high-pressure chamber and decompression, a specimen is loaded into the upper air space of the graduated cylinder, as shown in Figure 3. The main measurement system consists of a high-pressure chamber for gas exposure and a graduated cylinder immersed partially in a water container. Figure 3 illustrates the two volumetric methods employed to observe the water level: one method uses a digital camera, as shown in Figure 3b, and the other uses a capacitance meter, as shown in Figure 3c.

In the method shown in Figure 3b using a digital camera, the gas emitted from the rubber specimen leads to a reduction in the water volume in the graduated cylinder over time. The position of the water level is measured by the digital camera. The volume (*V*) and pressure (*P*) of the gas in the cylinder vary by emitted gas over time. The gas in the cylinder follows the ideal gas equation, *PV* = *nRT*, where R is the gas constant, 8.20544 × 10^−5^ m^3^·atm/(mol·K), *T* is the absolute temperature of the gas in the upper part of the cylinder, and *n* is the mole number of the emitted gas in the cylinder. The time-dependent P(t) and V(t) of the gas in the cylinder are formulated as [27,29,31]:(2)$$P(t)=P_{o}-\rho gh(t),\;V(t)=V_{o}-V_{s}-V_{h}(t)$$
where $P_{o}$ is the atmospheric pressure outside the cylinder, ρ is the density of the distilled water, *g* is gravitational acceleration, h(t) is the level (height) of the water volume in the cylinder based on the water level in the water container, $V_{o}$ is the total volume of both water and gas in the cylinder based on the water level in the water container, the time-varying $V_{h}(t)$ is the water volume in the cylinder measured based on the water level in the water container, and $V_{s}$ is the specimen volume.

The concentration of the gas emitted from the specimen is obtained by measuring the water volume $\left[V_{h}\left(t\right)\right]$ over time. Thus, the total mole number [n(t)] of the emitted gas is obtained by measuring the total gas volume [V(t)] in the cylinder, i.e., the reduction in the water volume.
(3)$$n(t)=\frac{P\left(t\right)V(t)}{RT(t)}=\frac{P\left(t\right)\left[V_{a}+V_{g}(t)\right]}{RT(t)}=\frac{P_{0}\left[1+\beta (t)\right]\left[V_{a}+V_{g}(t)\right]}{RT_{0}\left[1+\alpha (t)\right]}\;≅\frac{P_{0}}{RT_{0}}\left[V_{a}+V_{g}(t)+V\left(t\right)\left(\beta (t)-\alpha (t)\right)\right]=n_{a}\left(t\right)+n_{g}\left(t\right),$$
$${with\;n}_{a}\left(t\right)=\frac{P_{0}}{RT_{0}}V_{a},\;n_{g}\left(t\right)=\frac{P_{0}}{RT_{0}}\left[V_{g}(t)+V\left(t\right)\left(\beta (t)-\alpha (t)\right)\right]$$
$$\alpha (t)=\frac{T(t)-T_{0}}{T_{0}},\;\beta (t)=\frac{P(t)-P_{0}}{P_{0}}$$
where $T_{0\;}$ and $P_{0}$ are the initial temperature and pressure of the gas inside the cylinder, respectively, V(t) is the sum of the initial remaining air volume ($V_{a})$ and the emitted gas volume ${[V}_{g}\left(t\right)]$, i.e., $V\left(t\right)=V_{a}+V_{g}\left(t\right)$, $n_{a}$ is the initial air mole number, and $n_{g}(t)$ is the time-varying gas mole number corresponding to the gas volume increase by the emitted gas. Thus, $n_{g}(t)$ can be used to determine the emitted gas concentration $\left[C\left(t\right)\right]$ per mass from the rubber as follows [32]:(4)$$C\left(t\right)\left[\mathrm{wt}\cdot \mathrm{ppm}\right]=n_{g}(t)\left[mol\right]\times \frac{m_{g}\left[\frac{g}{mol}\right]}{m_{sample}\left[g\right]}\times 10^{6}\;=\frac{P_{0}}{RT_{0}}\left[V_{g}(t)+V\left(t\right)\left(\beta (t)-\alpha (t)\right)\right]\left[mol\right]\times \frac{m_{g}\left[\frac{g}{mol}\right]}{m_{sample}\left[g\right]}\times 10^{6}$$
where $m_{g}$ [g/mol] is the molar mass of the gas, for instance, for H_2_, $m_{H2}$ [g/mol] = 2.016 g/mol, and $m_{sample}$ is the specimen mass. According to Equations (3) and (4), the time-dependent gas mole number, $n_{g}(t)$, can be used to obtain the gas mass concentration, $[C\left(t\right)],$ by multiplying, $k=\left[\frac{m_{g}\;}{m_{sample}}\right].$ $n_{g}(t)$ and $C\left(t\right)$ are dependent on the variations in pressure and temperature. Thus, the variations in temperature and pressure are compensated for, leading to precise measurements.

We can also obtain $C\left(t\right)$ in emitted gas through the capacitance measurement to determine the water level (Figure 3c). A capacitor made with two semicylindrical electrodes mounted to the outer face of an acrylic tube is employed, as shown in Figure 4 [32]. The inner part of the acrylic tube is filled with a mixture of water and gas. The electrodes attached to the outer part of the acrylic tube are fabricated with a thin copper material. The capacitance in the acrylic tube depends on the dielectric permittivity of the media between the two capacitive electrodes. The dielectric permittivity of distilled water is approximately 78 times larger than that of the gas in the cylinder. Thus, the change in the water level in the two capacitor electrodes causes an appreciable change in the capacitance. Consequently, we measure the change in the actual capacitance caused by the changed water level. Thus, the water level corresponding to the measured capacitance is determined by the precalibration equation between the measured capacitance and water level [32]. The obtained water level can be used to determine the emitted gas mole number and gas concentration per mass according to Equations (3) and (4).

### 3.3. Manometric Measurement of the Gas Emitted by Enriched Specimens

Figure 5 illustrates the MM to measure the content of the released gas at 298 K; this apparatus consists of a high-pressure chamber for gas exposure and a cylindrical specimen container with a USB-type pressure/temperature logger and rubber seal.

After exposure under high pressure and decompression in the chamber, the specimen is moved into the cylindrical container, as depicted in Figure 5. The time lag between decompression and the start of the measurement is determined. The gas emitted from the specimen increases the pressure in the specimen container over time. Thus, the pressure [P(t)] and temperature $\left[T\left(t\right)\right]$ of the gas inside the sample container varies with time. The gas in the container follows the ideal gas equation, *PV* = *nRT*, where *n* is the mole number of the released gas in the specimen container.

The gas released from the specimen is obtained by measuring the increase in pressure $\left[P\left(t\right)\right]$ over time with the manometric measurement method at constant volume in the container. Thus, the total number of moles [n(t)] can be obtained by measuring the increase in the gas pressure [P(t)] due to the released gas in the cylindrical container as follows:(5)$$n(t)=\frac{P\left(t\right)V_{0}}{RT(t)}=\frac{P\left(t\right)V_{0}}{RT(t)}=\frac{{[P}_{0}{+\Delta P(t)]V}_{0}}{RT_{0}\left[1+\alpha (t)\right]}\;≅\frac{P_{0}V_{0}+\Delta P(t)V_{0}}{RT_{0}}\left[1-\alpha (t)\right]=n_{0}+\Delta n\left(t\right),$$
$${with\;n}_{0}=\frac{P_{0}V_{0}}{RT_{0}},\;\Delta n(t)=\frac{V_{0}}{RT_{0}}[\Delta P(t)-\alpha \left(t\right)P_{0}-\alpha \left(t\right)\Delta P(t)]$$
$$\alpha (t)=\frac{T(t)-T_{0}}{T_{0}}$$
where $T_{0}$, $V_{0}$ and $P_{0}$ are the initial temperature, initial air volume and initial pressure of the gas inside the cylindrical sample container, respectively, P(t) is the sum of the initial air pressure $\left(P_{o}\right)$ and time-varying released gas pressure $\left[\Delta P\left(t\right)\right]$ from the specimen, i.e., $P\left(t\right)=P_{o}+\Delta P\left(t\right)$, $n_{0}$ is the initial air mole number, and Δn(t) is the time-varying gas mole number corresponding to the increase in gas pressure due to the released gas. α(t) is the change rate of the temperature with regard to the initial temperature. Thus, Δn(t) can be transformed into the released gas concentration $\left[\Delta C\left(t\right)\right]$ performance mass for the polymeric specimen as:(6)$$\Delta C\left(t\right)\left[\mathrm{wt}\cdot \mathrm{ppm}\right]=\Delta n(t)\left[mol\right]\times \frac{m_{g}\left[\frac{g}{mol}\right]}{m_{specimen}\left[g\right]}\times 10^{6}\;=\frac{V_{0}}{RT_{0}}[\Delta P(t)-\alpha \left(t\right)P_{0}-\alpha \left(t\right)\Delta P(t)]\left[mol\right]\times \frac{m_{g}\left[\frac{g}{mol}\right]}{m_{specimen}\left[g\right]}\times 10^{6}$$
where $m_{g}$ [g/mol] is the molar mass of the gas used, for instance, for H_2_, $m_{H2}$ [g/mol] = 2.016 g/mol; for N_2_, $m_{N2}$ [g/mol] = 28.001 g/mol; and $m_{specimen}$ is the mass of the specimen.

The first term, ΔP(t), in Equation (6) is the pressure increase due to the gas emitted by the specimen. Two terms, $\left[-\alpha \left(t\right)P_{0}-\alpha \left(t\right)\Delta P(t)\right]$, in Equation (6) indicate the pressure change caused by temperature variation $\alpha \left(t\right)$, regardless of the emitted gas. According to Equations (5) and (6), the time-dependent gas mole number, Δn(t), can be transformed into the gas mass concentration, $[\Delta C\left(t\right)],$ by multiplying by $k=\left[\frac{m_{g}\;}{m_{specimen}}\right].$ Δn(t) and $\Delta C\left(t\right)$ are influenced by temperature variations. Thus, we must compensate for variations caused by temperature changes to obtain precise measurements. The changes in the gas volume and pressure due to the released gas are monitored based on the pressure, which is measured with the pressure data logger in the container vessel, as shown in Figure 5.

### 3.4. Gas Chromatography of the Gas Emitted by Enriched Specimens

To measure the concentration of the emitted gas, a GC method is applied, as shown in Figure 6. The pressure in the high-pressure chamber was reduced to atmospheric pressure by opening a needle valve, and the specimen was loaded into a quartz tube connected to a GC injector. The time lag between decompression and the start of the GC measurement was approximately 7 min.

The TDA–GC method quantitatively and qualitatively analyzes the testing gas by measuring the position and area of the identified separated GC signal [24], as shown in Figure 6b. The flow rate of the carrier gas (helium) is controlled using a mass flow controller. The gas emitted from the specimen is mixed with the helium carrier gas and sent to the capillary GC column through the injector. Then a PDD produces electrical signals corresponding to the separated gas components. Oxygen and nitrogen signals are not released by the specimen but are temporarily observed initially because of contact with air as the specimen is moved from the high-pressure chamber to the quartz tube, as shown in Figure 6b. TDA–GC has been set up for selectively H_2_ gas measurement. Thus, we have only provided the H_2_ results later. Diffusion analysis is applied based on the H_2_ peak released from the specimen.

Under the experimental conditions for the GC method at 1 atm, 298 K, and a helium carrier gas flow rate of 1.67 × 10^−7^ m^3^/s, the mass concentration of H_2_ per second, $C_{\mathrm{mass}}\left(\frac{\mathrm{wt}\cdot \mathrm{ppm}}{s}\right)$, can be determined according to Equation (7) [27],
(7)$$C_{mass}\left(\frac{\mathrm{wt}\cdot \mathrm{ppm}}{s}\right)≅1.392\times 10^{-5}\frac{C_{mol}(mol\cdot \mathrm{ppm})}{m_{sample}(g)}$$
where *C_mol_* (mol·ppm) is the molar concentration of H_2_ in ppm obtained from GC measurements, $m_{\mathrm{sample}}$ is the mass of the specimen, and the molar mass of H_2_ is 2.016 g/mol.

### 3.5. Diffusion Analysis Program for Obtaining Diffusion Parameters and its Algorithm

Assuming that the gas emitted from the gas-enriched specimens follows Fickian diffusion, the concentration $C_{E}(t)$ of the released gas in emission mode is formulated as [33,34]:(8)$$\begin{array}{ll} C_{E}(t)/C_{\infty } & =1-\frac{32}{\pi^{2}}\times \left[\sum_{n=0}^{\infty }\frac{exp\left\{\frac{{-\left(2n+1\right)}^{2}\pi^{2}Dt}{l^{2}}\right\}}{{\left(2n+1\right)}^{2}}\right]\times \left[\sum_{n=1}^{\infty }\frac{exp\left\{-\frac{D\beta_{n}^{2}t}{\rho^{2}}\right\}}{\beta_{n}^{2}}\right]\\ & \begin{array}{ll} =1-\frac{32}{\pi^{2}} & \times \left[\frac{\exp\left(-\frac{\pi^{2}Dt}{l^{2}}\right)}{1^{2}}+\frac{\exp\left(-\frac{3^{2}\pi^{2}Dt}{l^{2}}\right)}{3^{2}}+\ldots ,+\frac{\exp\left(-\frac{{\left(2n+1\right)}^{2}\pi^{2}Dt}{l^{2}}\right)}{(2{n+1)}^{2}}+\ldots ,\right]\\ & \times \left[\frac{\exp\left(-\frac{D\beta_{1}^{2}t}{\rho^{2}}\right)}{\beta_{1}^{2}}+\frac{\exp\left(-\frac{D\beta_{2}^{2}t}{\rho^{2}}\right)}{\beta_{2}^{2}}+\ldots ,+\frac{\exp\left(-\frac{D\beta_{n}^{2}t}{\rho^{2}}\right)}{\beta_{n}^{2}}+\ldots ,\right] \end{array} \end{array}$$
where $\beta_{n}$ is the root of the zeroth-order Bessel function J_0_ (β_n_). Equation (8) is the solution to Fick’s second diffusion law in the case of a cylindrical specimen. Equation (8) is an infinite series expansion consisting of two summations. The product of the two parentheses becomes $\frac{\pi^{2}}{32}$ at *t* = 0. Thus, the value $\frac{32}{\pi^{2}}$ is inserted in front of the equation so that the gas concentration at *t* = 0 becomes *C*_E_ (*t* = 0) = 0. Moreover, *C*_E_ (*t* = ∞) = $C_{\infty }$ is the total gas uptake obtained at infinite time. *D* is the gas diffusion coefficient, and l and ρ are the thickness and radius of the cylindrical specimen, respectively. In the remaining mode, Equation (8) can also be applied for the cylindrical specimen. The total gas uptake is defined as *C*_0_ at *t* = 0.

The residual concentration *c*(*t*) of the gas in the remaining mode for the spherical specimen is expressed as follows [29,33].
(9)$$C_{R}\left(t\right)=\frac{6}{\pi^{2}}c_{0}\sum_{n=1}^{\infty }\frac{1}{n^{2}}exp(-\frac{Dn^{2}\pi^{2}t}{a^{2}})$$

Equation (9) is the solution of Fick’s second diffusion equation for a spherical specimen. a is the radius of the spherical specimen and D is the diffusion coefficient. *C*_0_ is the total content of emitted H_2_ at *t* = 0. In Equations (8) and (9), more summation terms are needed to determine precise values for D and *C*_0_. Thus, we developed a dedicated diffusion analysis program using Visual Studio, which allows us to precisely calculate *D* and *C*_0_, with up to 50 terms included in the summation in Equations (8) and (9).

We developed a diffusion analysis program using a nonlinear optimization algorithm [35]. Figure 7 shows the flowchart of the diffusion analysis program developed to precisely analyze the $C_{E}\left(t\right)$ and $C_{R}(t)$ data using Equations (8) and (9), respectively, with the Nelder–Mead simplex nonlinear optimization algorithm [18,24]. The data type (remaining, emission, and transmission mode) and specimen shape (cylinder, sphere, and sheet) are used to determine the appropriate diffusion equation. The algorithm analyzes the gas uptake and diffusivity according to the solutions of the diffusion equation of Equations (8) or (9), and we choose the appropriate diffusion model corresponding to the specimen shape and the number of superposition models.

The diffusion analysis program is applied to a cylindrical-shaped specimen in the remaining mode, and the GM is used to obtain the *C*_0_ and *D* values of H_2_, as shown in Figure 8. Figure 8 shows the representative analytical results using the analysis program for a cylindrical NBR specimen (radius of 5.0 mm and height of 2.0 mm) exposed to a pressure of 35 MPa. The *D* and *C*_0_ values are obtained by substituting the remaining relative H_2_ mass (*C_R_*) at each time from Equation (1) into Equation (8) and optimizing each parameter with the least squares method. Finally, we determine that *D* = 1.86 × 10^−11^ m^2^/s and *C*_0_ = 986 wt∙ppm. The red line and × symbol indicate the fit obtained using Equation (8) and the experimental data, respectively. The arrow in the unknown parameter list indicates the diffusion coefficient, *D*, obtained from the diffusion analysis program. The figure of merit (FOM) of 3.3% is the standard deviation between the measured data and Equation (8). The arrow on the y-axis in Figure 8 corresponds to the value of *C*_0_ obtained at *t* = 0 by extrapolating the fitted line.

## 4. Results and Discussion

We demonstrate the representative results in the four methods based on uptake and diffusivity obtained by applying diffusion analysis program.

### 4.1. Gravimetric Method

For the GM using an electronic balance (Figure 2), the residual mass (*C_R_*) of the specimen based on the H_2_ emitted over time is obtained by measuring the change in mass of the specimen according to Equation (1). The measured results are shown in Figure 9. The diffusion analysis program is used to obtain the *C*_0_ and *D* values of H_2_ for spherical-shaped NBR specimens exposed to hydrogen at a high pressure of 10.2 MPa [27]. The *D* and *C*_0_ values are obtained by substituting the residual H_2_ concentration after the elapsed time determined from Equation (1) into Equation (9) and optimizing each parameter by the least squares method. Thus, the values *D* = 1.27 × 10^−10^ m^2^/s and *C*_0_ ≈ 696 wt∙ppm are determined by extrapolation. The dashed line and filled squares indicate the fitted line of Equation (9) and the experimental data, respectively. The arrow in Figure 9 corresponds to the value of *C*_0_ obtained at *t* = 0 by extrapolating the fitted line.

In addition, Figure 10 shows the representative result of H_2_ uptake content and diffusivity versus pressure for the spherical NBR, EPDM, and FKM specimens determined by the GM [27]. The H_2_ uptake followed Henry’s law [36,37], as shown by the black dashed lines in Figure 10a. The obtained slopes for the NBR, EPDM, and FKM specimens are shown in Figure 10a. The determined diffusivity in Figure 10b does not show pressure-dependent behavior for the specimens. Thus, the average diffusivity is taken as a representative value for the investigated pressure. The average diffusivity of the NBR, EPDM and FKM specimens is determined, as shown by the black dashed horizontal line in Figure 10b. The error bars in (a) and (b) indicate the standard deviation of the measured value.

### 4.2. Volumetric Method

The emitted N_2_ gas content and its diffusivity are obtained through the VM using a graduated cylinder (Figure 3b). Figure 11 displays the two procedures for obtaining the N_2_ gas diffusion parameters in the EPDM sample with a digital camera [32]. Figure 11a shows the time-dependent water level (emitted gas volume) measured by a digital camera. Figure 11b depicts the time-dependent mass concentration determined according to the measured water level and Equation (4). The nitrogen emission is saturated 60,000 s after decompression.

By applying the diffusion analysis program results shown in Figure 8, *D* and *C_∞_* are obtained by fitting the emitted H_2_ content to Equation (8) and optimizing the parameters with the least squares method. The black line in Figure 11b indicates the line fitted with Equation (8) based on the experimental data. Thus, the value *D* = 5.35 × 10^−11^ m^2^/s is found. The blue solid line represents the total compensated emission curve used to restore the missing content caused by the time lag, and *C_∞_* = 3382 wt∙ppm is the value obtained at *t* = *∞* by extrapolating the fitted line.

On the other hand, Figure 12 shows the procedure for acquiring *D* and $C_{\infty }$ for the same EPDM cylindrical specimen by using two semicylindrical electrodes (Figure 3c and Figure 4) [32]. Figure 12a represents the precalibration result according to the second polynomial equation between the water level and the measured capacitance by quadratic regression. Figure 12b shows the water level determined according to the measured capacitance, where the black and blue squares indicate the measured capacitance and water level, respectively. Figure 12c shows the diffusion parameters, *D* and $C_{\infty }$, obtained using the diffusion analysis program and Equation (8). The blue line in Figure 12c represents the total compensated emission curve, including the missing content due to the lag time. The investigated results in Figure 11 and Figure 12 are consistent within the experimental uncertainty.

With the volumetric method using Figure 3b, we measure the emitted hydrogen content at pressures up to 90 MPa for one neat EPDM specimen and nine EPDM composites blended with fillers [38]. Figure 13 represents the H_2_ uptake versus pressure for four representative EPDM specimens. Figure 13a–d show the relationship between the pressure and H_2_ uptake for the neat EPDM, EPDM composites compounded with silica fillers (40 phr and 60 phr), EPDM HAF40, and EPDM SRF40 specimens, respectively.

The H_2_ uptake (*C*_∞_) for the neat EPDM and EPDM S20 (Figure 13a,b) linearly increases with increasing pressure up to 90 MPa, satisfying Henry’s law. This is attributed to H_2_ absorption into the polymer matrix. However, in Figure 13c,d, the H_2_ uptake in the EPDM HAF40 and SRF40 deviates from Henry’s law at pressures above 15 MPa; this behavior is responsible for the H_2_ adsorbed at the CB filter. Thus, dual sorption is observed for all CB-blended EPDM composites. The dual-mode sorption phenomenon is observed over the entire pressure range up to 90 MPa can be expressed as [36,37,39,40,41,42,43]:(10)$$C_{\infty }=kP+\frac{abP}{1+bP}$$

$C_{\infty }$ indicates total H_2_ gas uptake. The first term is related to Henry’s law and has a coefficient *k*. The second term indicates the Langmuir model with a representing the maximum adsorption quantity and b representing the adsorption equilibrium constant.

### 4.3. Manometric Method

The emitted O_2_ content is measured by the MM, as shown in Figure 5. The representative result for the O_2_ emitted over time is shown in Figure 14a for a cylindrical LDPE specimen at a pressure of 6.0 MPa. The O_2_ uptake and diffusivity are determined by applying the diffusion analysis program and Equation (8). In Figure 14a, single-mode emission/diffusion behaviors for LDPE plastic are observed based on the time-varying gas uptake data. The single-mode oxygen emission for the LDPE specimen is caused by O_2_ diffusion, which is attributed to the absorption of O_2_ in the amorphous phase. The emitted oxygen content is obtained from *t* = 6 min after decompression due to the time lag. Thus, the missing amount of oxygen emitted from *t* = 0 min to *t* = 6 min after decompression is determined with the diffusion analysis program. The missing content is included in the oxygen emission data, as shown by the blue line of Figure 14a.

The oxygen emitted by the LDPE specimen in Figure 14b linearly increases with increasing pressure, following Henry’s law. However, the O_2_ diffusion coefficient for the LDPE specimen in Figure 14c remains constant at 4.17 × 10^−11^ m^2^/s within the experimental uncertainty (± 0.5 × 10^−11^ m^2^/s), regardless of the pressure.

Single-mode emission and diffusion behaviors of N_2_ for the cylindrical LDPE plastic specimen are also observed in Figure 15a. The single-mode N_2_ emission for LDPE is caused by N_2_ diffusion. The nitrogen emitted by the LDPE specimen in Figure 15b is proportional to the pressure, following Henry’s law. Moreover, the nitrogen diffusivity for the LDPE specimen in Figure 15c remains constant at 2.56 × 10^−11^ m^2^/s, regardless of pressure. In summary, two pressure-dependent emission and diffusivity behaviors for O_2_ and N_2_ gases are very similar.

### 4.4. Gas Chromatography

The emitted hydrogen from specimen is measured by a GC system within the corresponding time interval, as shown in Figure 6. Figure 16a shows representative time-varying H_2_ concentration data determined according to the mass concentration per second based on the GC data and Equation (7) for EPDM specimens at a pressure of 4.85 MPa. The data in Figure 16a over time was used to obtain the saturation value, i.e., the amount of H_2_ absorbed by the EPDM specimen, as shown in Figure 16b. The extrapolated H_2_ concentration is 103 wt∙ppm, which is the saturated value at infinite time. The content is obtained from the time delay of 6 min. Thus, the missing amount of hydrogen released up to time delay after decompression was determined by applying the diffusion analysis program [24].

By subtracting the emission values (Figure 16b) from the exposed emission value (103 wt∙ppm), we obtain the remaining H_2_ concentration. Then, we obtain *D* and *C*_0_ through the diffusion analysis program [24]. The application window of the diffusion analysis program is shown in Figure 17a. The analysis results show two diffusion characteristics, i.e., slow and fast, and the experimental data fit the sum of these two components, as shown in Figure 17b. *D* and *C*_0_ are as follows: *D*_fast_ = 3.51 × 10^−10^ m^2^/s, *C*_0-fast_ = 161 wt∙ppm, *D*_slow_ = 2.64 × 10^−11^ m^2^/s, and *C*_0-slow_ = 43.8 wt∙ppm. After the cylindrical EPDM specimen was exposed to H_2_ at 4.85 MPa, we determine (*C*_0_ = 204.7 wt∙ppm) the total H_2_ uptake. The total uptake value, *C*_0_, is the value on the y-axis at *t* = 0, which is obtained by extrapolating the fitted line with the diffusion analysis program.

## 5. Comparisons of the Measured Results and Characteristics of the Four Methods

The developed methods for measuring H_2_ uptake and diffusivity are validated by comparing the measured values for the same specimen. The H_2_ uptake, solubility and diffusivity results obtained with the GM, VM, MM and GC method are consistent within the experimental uncertainty, as shown in Figure 18 [27]. The solubility is obtained from the slope of the uptake data with pressure shown in Figure 10a. The uncertainty estimations for solubility and diffusivity are well described in prior investigations [24,26]. The relatively large uncertainty in the GM is caused by variations due to temperature, humidity and instability of the electronic balances.

Based on the measured results and analysis, we review the performance of the four developed methods. The characteristics of the measuring methods are presented in terms of measurand, resolution, stability, uncertainty, figure of merit (FOM), sensitivity, and features in Table 1. The resolutions, indicating the lowest readable digits for the four methods, were obtained from the specifications of the equipment and measurand for each method. Compared with the other methods, the lowest resolution of 0.01 wt∙ppm was estimated for the GC method, corresponding to the area of the smallest hydrogen GC peak obtained from the GC signal array versus time after decompression. The stability can be represented as the standard deviation obtained by repeated measurements for 24 h after the H_2_ emission is finished, which were approximately 0.13~0.22%. The expanded uncertainty indicates the uncertainty in the diffusivity measurement, represented as the product of the combined standard uncertainty and coverage factor. The expanded uncertainties for the VM, MM, and GC method are less than 10%, but the expanded uncertainty for the GM is higher. The relatively large uncertainty for the GM is due to the large uncertainty in the temperature/humidity, uncertainty in the electronic balance measurements and standard deviation of the fitting result. The FOM is the standard deviation between the measured data and the fits from Equations (8) and (9). A FOM value less than 1%, as shown by three methods, implies good agreement between the measured and theoretical values. The sensitivity to temperature and pressure of the methods is also presented in Table 1.

An electronic balance detects minute-level changes in electric resistance, proportional to the deforming force acting by the principle of the Wheatstone bridge. Thus, the GM is a sensitive technique that is influenced by the offset of the electronic balances and the stability of the temperature and humidity. Although the GM is simple and inexpensive, the electronic balance for GM must be operated in a controlled environment that maintains both temperature and humidity.

The VM using a graduated cylinder is an effective approach for determining gas transport parameters. This simple method could be used for on-site evaluations of gas diffusion because of the simple equipment employed. The VM for detecting the water level can be applied in real time by using a digital camera or capacitance meter with two capacitor electrodes, which requires precalibration between capacitance and water level prior to the measurements.

On the other hand, the MM is also simple, inexpensive, and effective. This method is more appropriate for on-site evaluations in the field than the VM. However, this method is very sensitive to temperature and thus requires pressure compensation to address temperature variations.

GC is a sophisticated method with complicated procedures for obtaining H_2_ uptake and diffusivity data according to individual GC peaks. This technique could be used to quantitatively evaluate the H_2_ content in small samples because of its good resolution. This sensitive method is the best candidate for precise analysis of H_2_ gas properties with multiple gas components. In summary, the main features of the four methods are described in the last line of Table 1.

## 6. Conclusions

Four methods for measuring gas uptake and diffusivity in polymers enriched with gas under high pressure up to 90 MPa are developed. These methods are based on measurements of the mass, volume, and pressure of the emitted gas and include the gravimetric, volumetric, manometric, and gas chromatography methods. The representative results of gas uptake, solubility, and diffusivity obtained from four methods are demonstrated in the several polymer specimens. In addition, the pressure-dependent gas uptake is associated with Henry’s law or Langmuir model depending on the polymer characteristics. The gas uptake and diffusivity obtained by these methods are consistent, and the characteristics and performance are reviewed. The features of these methods are presented as follows:The three developed methods are inexpensive and simple techniques, except for gas chromatography, for evaluating the gas uptake and diffusion of gas-enriched polymer materials under high-pressure conditions.The four developed methods are insensitive to variations in temperature and pressure, regardless of the specimen size, specimen shape, and testing gas species.All the methods are exactly calculable by applying the diffusion analysis program, which includes more than 50 terms in the summation of provided equation.The volumetric and manometric methods are flexible techniques in which the sensitivity, resolution, and range can be flexibly changed.All the methods are independent techniques, without any interaction between the testing gas molecule and gas sensor.The volumetric method is visible because the entire process of gas release and leakage can be observed by monitoring the change in water level.

The proposed techniques for effectively measuring the transport property of gas in sealing elastomers have different characteristics. Thus, the complementary four methods could be utilized in the testing of permeation properties of rubber material and O-ring under high pressure for hydrogen fueling stations.

## Figures and Tables

**Figure 1 polymers-16-00723-f001:**
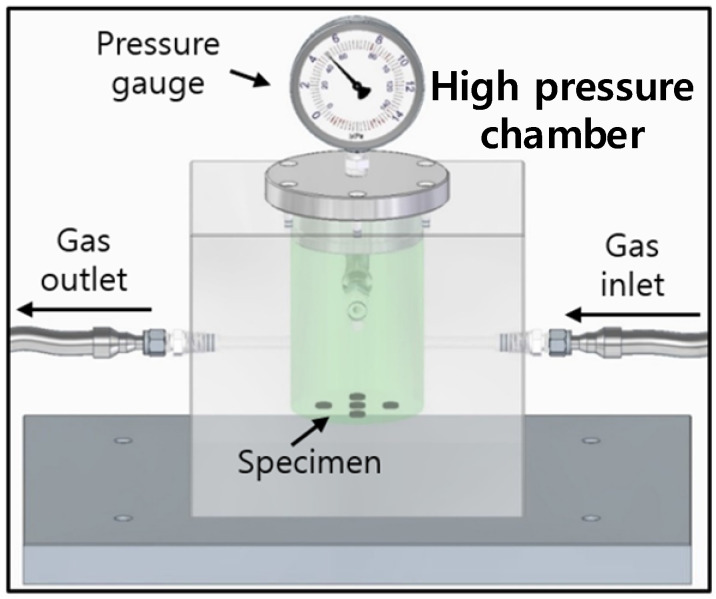
High-pressure chamber and gas exposure of specimens under high pressure conditions. The light gray rectangular box indicates the main body containing the cylindrical high-pressure chamber manufactured from SUS 316 material to withstand pressures up to 100 MPa, and the dark gray box below the chamber represents the shelf plate on which the chamber was horizontally placed during gas exposure.

**Figure 2 polymers-16-00723-f002:**
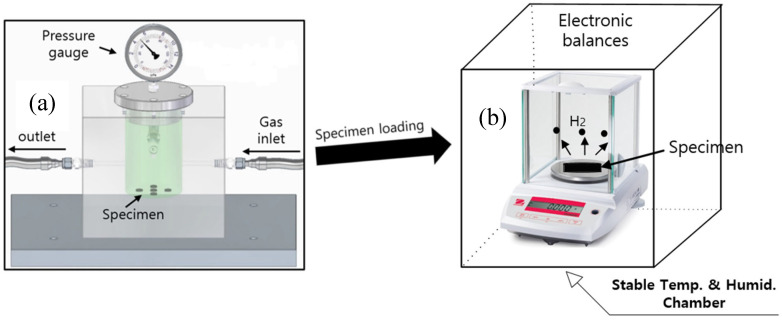
Gravimetric measurement of emitted gas employing an electric balance after gas exposure in a high-pressure chamber and subsequent decompression: (**a**) Specimen exposed to gas in a high-pressure chamber; (**b**) Real-time gravimetric measurement of the specimen by an electronic balance in a chamber with stable temperature and low humidity. The symbol (●) in (**b**) indicates the hydrogen gas emitted from the specimen.

**Figure 3 polymers-16-00723-f003:**
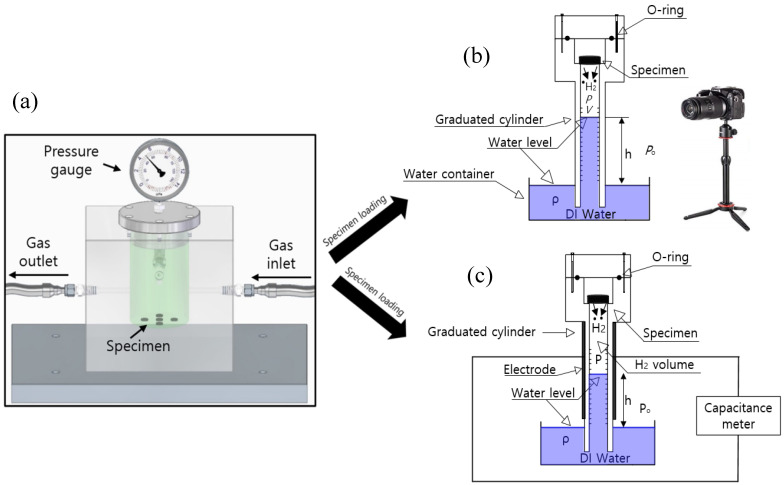
Volumetric measurement of gas concentration using a graduated cylinder after high-pressure exposure and subsequent decompression: (**a**) Specimen exposed in a high-pressure chamber. After chamber decompression and loading the specimen in the cylinder, the emitted gas was measured (water level measurement) by (**b**) a digital camera and (**c**) a capacitance meter employing a frequency response analyzer. Blue indicates distilled water. The symbol (●) in (**b**,**c**) indicates the testing gas emitted from the specimen.

**Figure 4 polymers-16-00723-f004:**
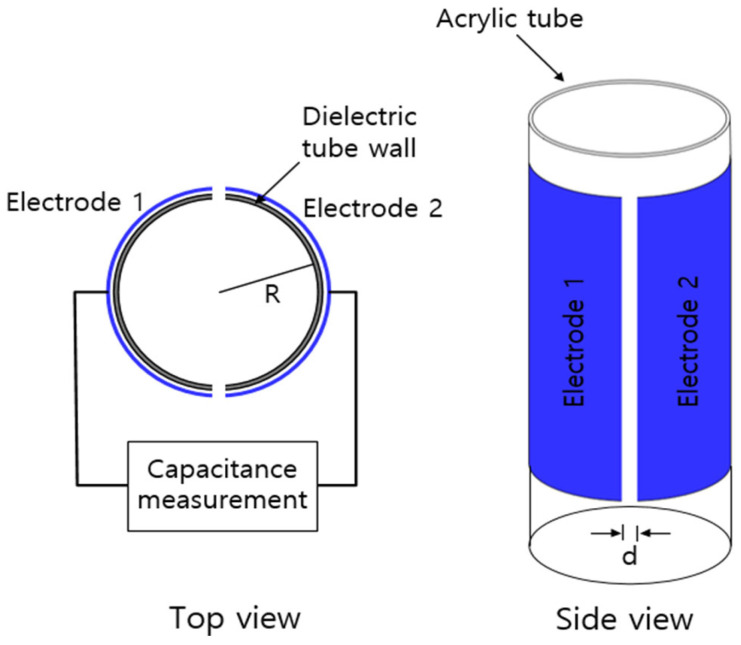
Top and side views for two semicylindrical capacitor electrodes, indicated in blue.

**Figure 5 polymers-16-00723-f005:**
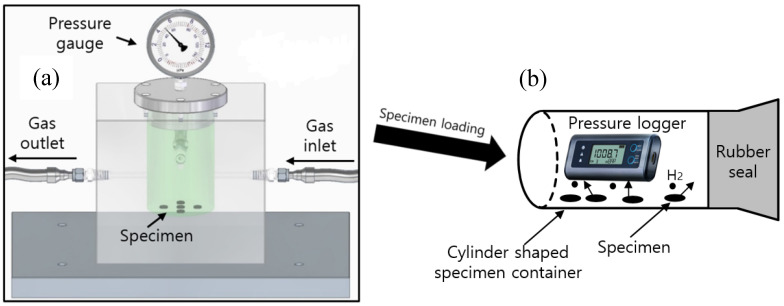
Manometric method for measuring gas uptake and diffusivity of the enriched specimen by employing a pressure/temperature logger: (**a**) Specimen exposed to gas (gas enriched) in a high-pressure chamber; (**b**) After chamber decompression, the enriched specimen is loaded in the cylindrical container. The emitted gas content is measured by a pressure logger in the specimen container. The symbol (●) in (**b**) indicates the testing gas emitted from the specimen.

**Figure 6 polymers-16-00723-f006:**
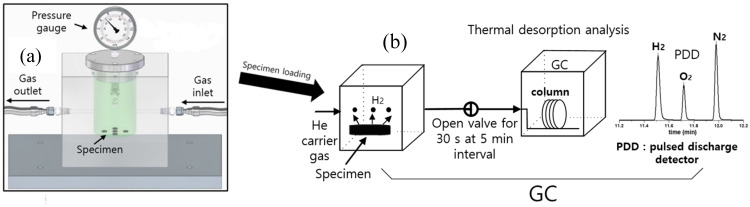
Gas chromatography method for measuring the emitted gas concentration and diffusivity: (**a**) Specimen exposed in the chamber under high pressure; (**b**) Gas chromatography measurement after decompression. The symbol (●) in (**b**) indicates hydrogen gas emitted from the specimen.

**Figure 7 polymers-16-00723-f007:**
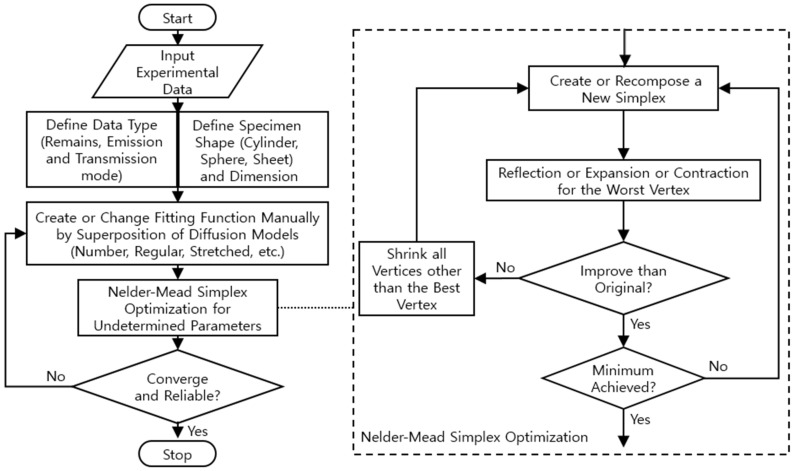
Flowchart for analyzing the gas uptake and diffusivity of specimens with different data types and various shapes with the Nelder–Mead simplex nonlinear optimization algorithm.

**Figure 8 polymers-16-00723-f008:**
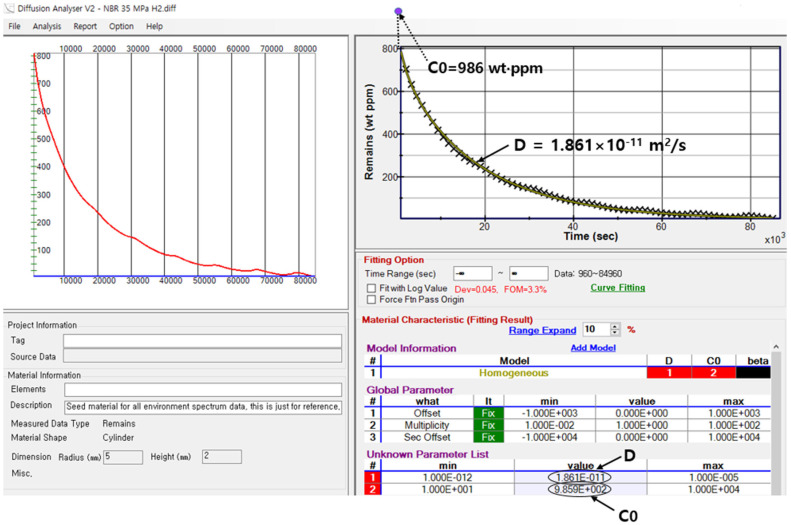
H_2_ remaining mass concentration versus time for a cylindrical NBR specimen exposed to a high pressure of 35 MPa. The fitting results of the total gas uptake and diffusivity (*C*_0_ and *D*) by the diffusion analysis program are shown in the right side.

**Figure 9 polymers-16-00723-f009:**
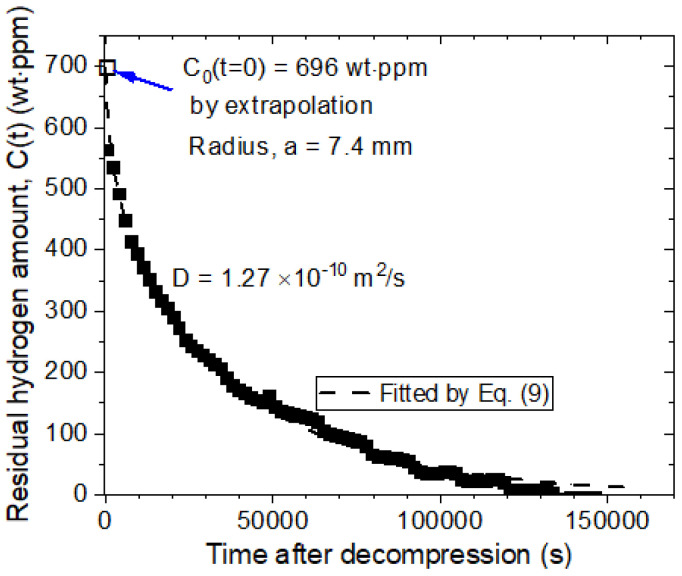
Residual H_2_ mass versus time after decompression obtained by the diffusion analysis program and Equation (9) for a spherical NBR specimen exposed to a pressure of 10.2 MPa. The emitted H_2_ was saturated after *t* = 150,000 s.

**Figure 10 polymers-16-00723-f010:**
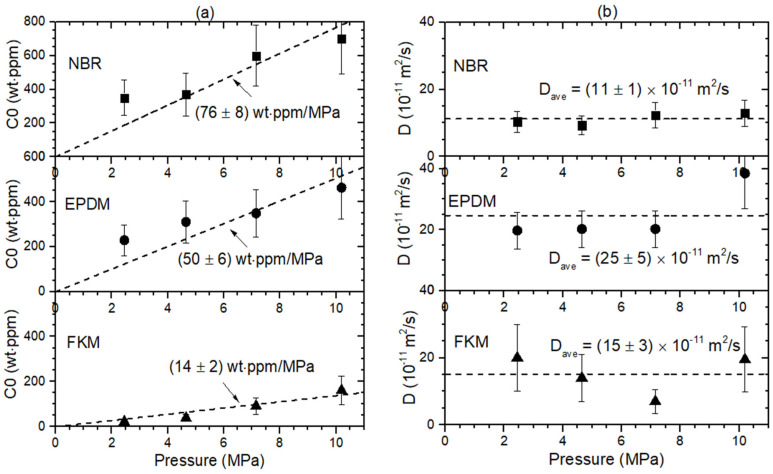
Pressure-dependent hydrogen uptake and diffusivity behaviors. (**a**) H_2_ uptake (*C*_0_) and (**b**) diffusivity (*D*) versus pressure for spherical NBR, EPDM and FKM with a radius of 10.0 mm obtained by GM.

**Figure 11 polymers-16-00723-f011:**
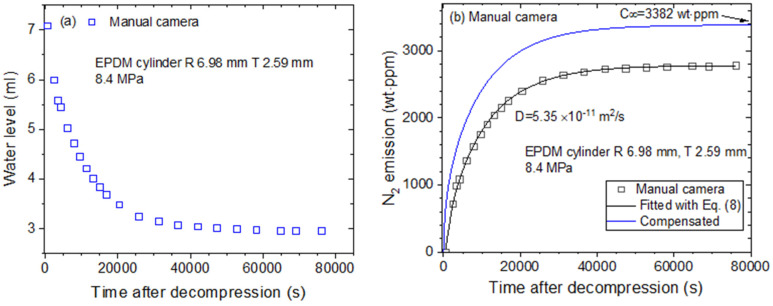
The procedure for obtaining the diffusion parameters in cylindrical EPDM specimens by using a digital camera: (**a**) Time-dependent water level versus time after decompression; (**b**) N_2_ emission content versus time. *D* and $C_{\infty }$ are obtained using the diffusion analysis program. *R* is the radius, and *T* is the thickness of the cylindrical-shaped elastomer.

**Figure 12 polymers-16-00723-f012:**
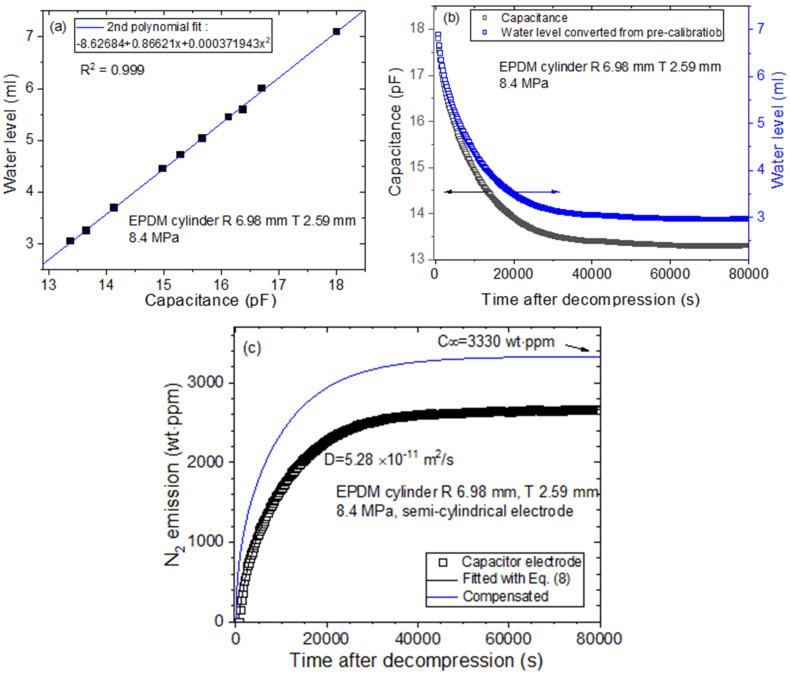
The procedures for acquiring the diffusion parameters with the cylindrical EPDM specimens by employing two semicylindrical capacitor electrodes and a frequency response analyzer as a capacitance meter: (**a**) Precalibration result with a second polynomial equation between the water level and capacitance; (**b**) Water level determined according to the measured capacitance, where the black and blue squares correspond to the measured capacitance and water level, respectively; (**c**) Diffusion parameters, *D* and $C_{\infty }$, obtained using the diffusion analysis program and Equation (8). R is the radius, and T is the thickness of the cylindrical-shaped elastomer.

**Figure 13 polymers-16-00723-f013:**
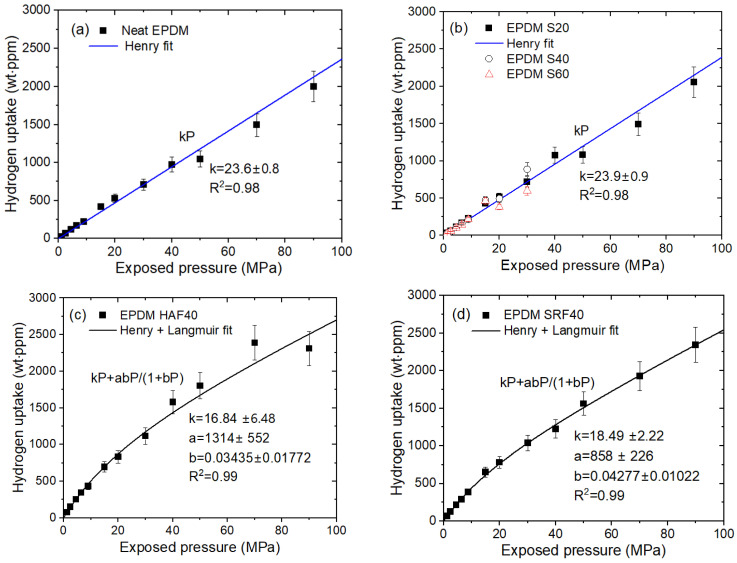
H_2_ uptake ($C_{\infty }$) versus pressure for (**a**) neat EPDM, (**b**) EPDM S40 and EPDM S60, (**c**) EPDM HAF40 and (**d**) EPDM SRF40. The blue and black solid lines represent the Henry and dual-mode (Henry + Langmuir) fits, respectively, showing the linear least squares fitting plots with squared correlation coefficients (*R*^2^). Cylindrical elastomers with a diameter of 13 mm and thickness of 3 mm were used. EPDM S20 indicates EPDM blended with silica of 20 phr (parts per hundred parts of rubber). EPDM HAF40 and EPDM SRF40 indicate EPDM blended with HAF (high abrasion furnace) and SRF (semi reinforcing furnace), respectively, carbon black filler of 40 phr.

**Figure 14 polymers-16-00723-f014:**
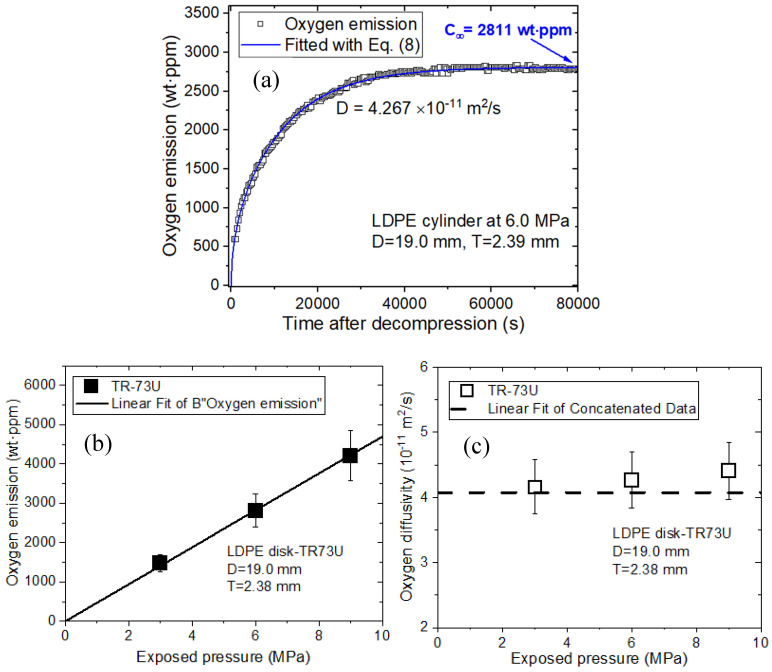
Single-mode O_2_ diffusion and desorption behaviors determined by the manometric method for LDPE plastic specimens. (**a**) O_2_ emission versus time at 6.0 MPa and subsequent decompression. The blue line in (**a**) is fitted with Equation (8). Total oxygen uptake (*C*_∞_), including missing content, is indicated by the blue arrow. (**b**) Linear relationship between the oxygen emission and pressure. The black line represents the linear Henry’s law fit. (**c**) Oxygen diffusivity versus pressure. The horizontal dashed line in (**c**) indicates the average diffusivity. TR-73 U in (**b**,**c**) indicates the manometer type for measuring temperature and pressure. *D* is the diameter, and *T* is the thickness of the cylindrical specimen.

**Figure 15 polymers-16-00723-f015:**
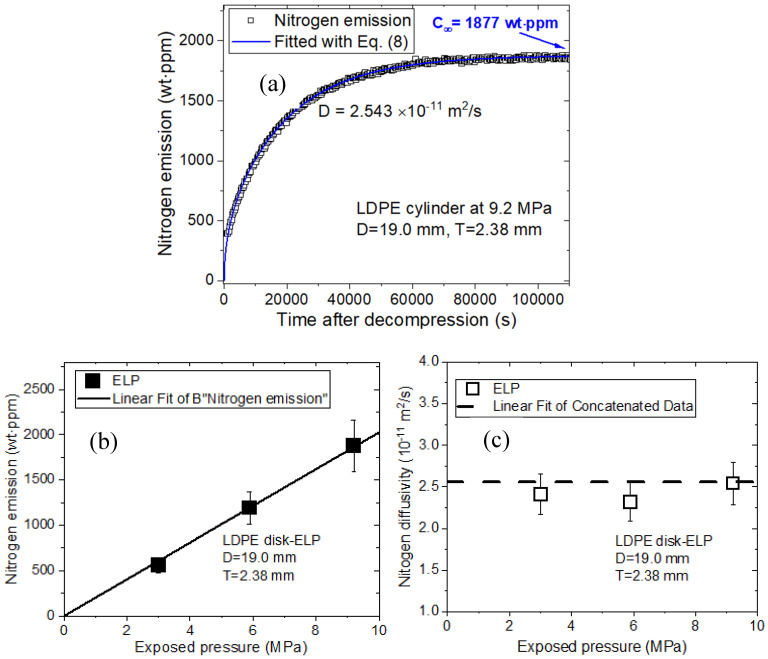
Single-mode N_2_ diffusion and desorption behaviors by the manometric method in the LDPE specimen: (**a**) N_2_ emission versus time at 9.2 MPa after decompression. The blue line in (**a**) is fitted with Equation (3). Total nitrogen uptake (*C*_∞_), including missing content, is indicated by the blue arrow; (**b**) Linear relationship between the emitted nitrogen and pressure. The black line represents the linear fit determined based on Henry’s law; (**c**) Nitrogen diffusivity versus pressure. The horizontal dashed line in (**c**) indicates the average diffusivity. ELP in (**b**,**c**) indicates the manometer type for measuring temperature and pressure. *D* is the diameter, and *T* is the thickness of the cylindrical specimen.

**Figure 16 polymers-16-00723-f016:**
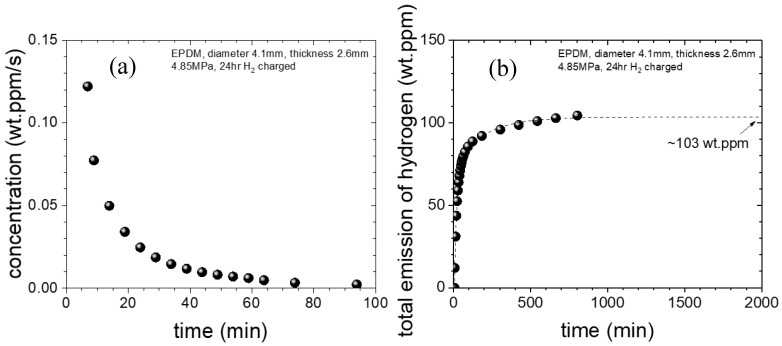
Time-varying H_2_ concentration for the EPDM specimen: (**a**) Remaining mass concentration of H_2_ per second from the GC data; (**b**) Time integration of the data in (**a**), which was used to obtain the saturation value, i.e., the total H_2_ content exposed to the EPDM specimen. The extrapolated total H_2_ content in (**b**) was 103 wt∙ppm.

**Figure 17 polymers-16-00723-f017:**
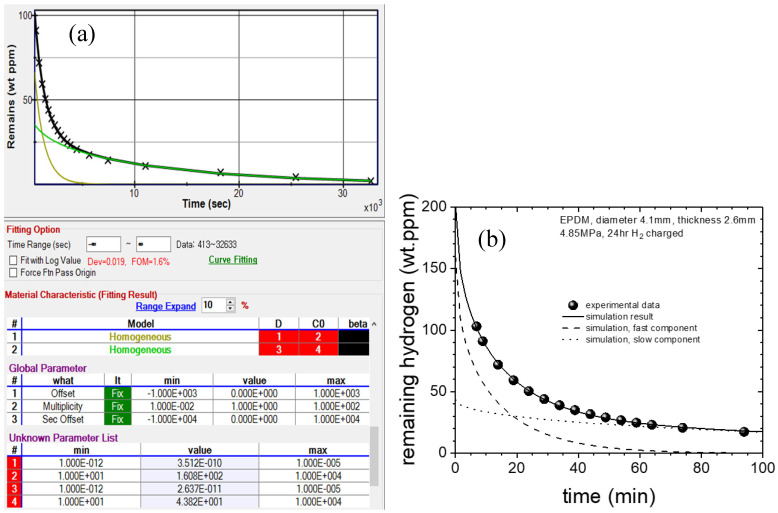
An example of remaining H_2_ concentration versus time for the EPDM specimen: (**a**) Application of the analysis program representing two diffusion behaviors. *D* and *C*_0_ are the diffusivity and the residual H_2_ concentration, respectively; (**b**) Experimental data of residual H_2_ concentration and simulation results indicated by solid lines, which are the sums of the dashed and dotted lines.

**Figure 18 polymers-16-00723-f018:**
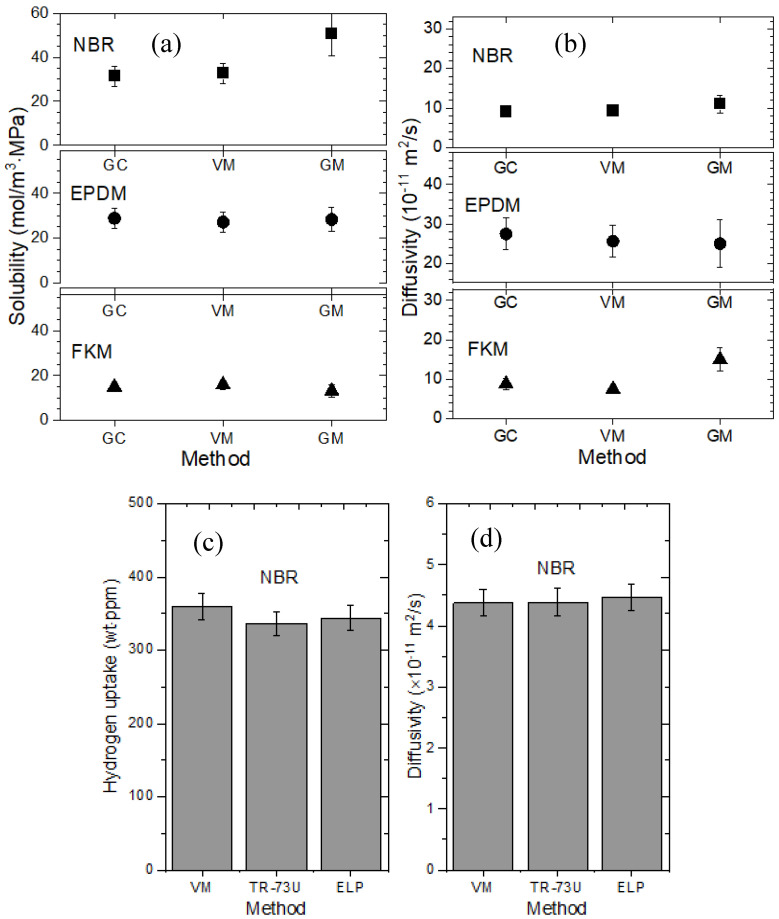
Comparisons between the GM, VM, MM, and GC methods for NBR, EPDM, and FKM specimens: (**a**) H_2_ solubility obtained by different methods for the NBR, EPDM, and FKM specimens; (**b**) H_2_ diffusivity obtained by different methods for the NBR, EPDM, and FKM specimens; (**c**) H_2_ uptake obtained by different methods for the NBR specimen, and (**d**) H_2_ diffusivity obtained by different methods for the NBR specimens. The TR-73 U and ELP in (**c**,**d**) indicate the different types of manometers used in the MM. Spherical specimens with a radius of 10.0 mm are used to obtain the data in (**a**,**b**). Cylindrical specimens with a radius of 7.46 mm and a thickness of 2.01 mm are used to obtain the data in (**c**,**d**).

**Table 1 polymers-16-00723-t001:** Comparison of the characteristics and performance of the four developed methods.

Measuring Methods	GM	VM	MM	GC
Measurand	mass	volume	pressure	volume
Resolution	0.3 wt∙ppm	0.1 wt∙ppm	0.1 wt∙ppm	0.01 wt∙ppm
Stability	<0.22%	<0.16%	<0.15%	<0.13%
Expanded uncertainty	<13%	<10%	<10%	<10%
Figure of merit (FOM)	1.5%	0.8%	0.7%	0.5%
Sensitivity to temperature and pressure	sensitive	sensitive	very sensitive	less sensitive
Features	very simple and inexpensive	simple, inexpensive and effective	very simple, inexpensive and effective	complicated and delicate

## Data Availability

The data used to support the findings of this study are available from the corresponding author upon request.

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
