# Peer review of "Review of Developed Methods for Measuring Gas Uptake and Diffusivity in Polymers Enriched by Pure Gas under High Pressure"

_polymers, 2024, doi:10.3390/polym16050723_

Round 1
Reviewer 1 Report
Comments and Suggestions for Authors
It is unclear whether this is a review or research article related to the gas diffusion measurement.
Below are my comments:
1. Title: If this review article is targeted towards hydrogen applications, it should be reflected in the title too. But based on the content, it includes other types of pure gases. Hence, strong justifications needs to be provided on why these gases are selected.
2. Abstract: “To accommodate the requirements of different environments, we firstly developed four complementary effective methods for measuring the gas absorption uptake from polymers enriched by pure gas under high pressure and determining the gas diffusivity.” Who are the ‘we’ in the statement? It is repeated in the content as well, but this is a single author work. Why there is development of methods stated in abstract when this is a review article?
3. What is the specific issue that the author would like to address? It is not clear from the introduction section.
4. There should be a dot in between “wtppm” unit in section 3.1.
5. What is the range of high pressure in this study? Justify why ideal gas equation is valid to be used in this study.
6. For GC, what’s the approximate peak time for He and Ar? Why only H2 are discussed in the section?
7. Some of the results shown are for H2, while some are for N2. It is very confusing to follow the flow of the results and why it is presented that way.
Author Response
Manuscript : polymers-2860165
Dear Editor and Reviewers,
Thank you for giving us the opportunity to submit a revised draft of the manuscript “Review of Developed Methods for Measuring Gas Uptake and Diffusivity in Polymers Enriched by Pure Gas under High Pressure” for publication in the polymers. We appreciate the time and effort that you and the reviewers dedicated to providing feedback on our manuscript and are grateful for the insightful comments on and valuable improvements to our manuscript. We have incorporated most of the comments and suggestions made by the reviewers. Those changes are represented with blue color within the manuscript for a point-by-point response to the reviewers’ comments and concerns. We reduced self-citations to be 11 % (final ref. number, 5) from 21 % by removing our five references (27, 29, 32, 35 and 42).
Reviewer 1
Comments and Suggestions for Authors :
It is unclear whether this is a review or research article related to the gas diffusion measurement.
Below are my comments:
- Title: If this review article is targeted towards hydrogen applications, it should be reflected in the title too. But based on the content, it includes other types of pure gases. Hence, strong justifications needs to be provided on why these gases are selected.
Author response: Thank you for valuable comments. This is the review article describing developed methods for obtaining the gas transport properties of various pure gases including H2, rather than targeting the applications of hydrogen infrastructure. In other words, we review the performance and characteristics of gases obtained by the GM, VM, MM and GC methods developed in previous investigations. We have also inserted the justifications why these gases are selected in the revised manuscript (page 2, line 71-78) as follows:
In this review article, the three types of gases (H2, N2 and O2) in the experiment and analysis were chosen as they were cheap and easily available. Although the review was not described, the previous research was related the diffusivity and solubility obtained from the experiments for the five gases (H2, He, N2, O2 and Ar) to the kinetic diameter and critical temperature, respectively. Thus, the five gases were selected in previous works because they have an appropriate wider range in the kinetic diameters and critical temperatures. However, we only demonstrate the results of three representative pure gases (H2, N2 and O2) in the review article.
- Abstract: “To accommodate the requirements of different environments, we firstly developed four complementary effective methods for measuring the gas absorption uptake from polymers enriched by pure gas under high pressure and determining the gas diffusivity.” Who are the ‘we’ in the statement? It is repeated in the content as well, but this is a single author work. Why there is development of methods stated in abstract when this is a review article?
Author response: Thank you for pointing this out. In response to reviewer comments, this is the review article for four methods developed previously by us. Although the manuscript has a single author, the developments were conducted by collaboration between our group and university members. Thus we have stated “we” in the original manuscript.
- What is the specific issue that the author would like to address? It is not clear from the introduction section.
Author response: Thank you for pointing this out. The review article describes the performance and characteristics in the measurement of various gases uptake and diffusivity obtained by developed methods. In the review work, we have also contained the comparisons among four methods together with features of these methods. In response to reviewer comments, we have complemented the clear statement in the introduction section of revised manuscript (page 2, line 71-83) as follows;
In this review article, the three types of gases (H2, N2 and O2) in the experiment and analysis were chosen as they were cheap and easily available. Although the review was not described, the previous research was related the diffusivity and solubility obtained from the experiments for the five gases (H2, He, N2, O2 and Ar) to the kinetic diameter and critical temperature, respectively. Thus, the five gases were selected in previous works because they have an appropriate wider range in the kinetic diameters and critical temperatures. However, we only demonstrate the results of three representative pure gases (H2, N2 and O2) in the review article. The measuring principles, measuring procedures, and representative results of the methods are described. We review the performance and characteristics of the GM, VM, MM and GC method developed in the previous researches. They included the investigations of various gases uptake and diffusivity obtained by developed methods. We have also contained the comparisons among four methods together with features of these methods.
- There should be a dot in between “wtppm” unit in section 3.1.
Author response: Thank you for pointing this out. In response to reviewer comments, we have inserted a dot between “wt and ppm” (wt∙ppm) in the revised manuscript as below;
page 2 line 96, Eq. (1), Eq. (4), Eq. (6), page 11 line 309, page 12 line 330, page 14 line 370, page 18 line 466,474, 476, 481, 482, page 20 line 514 and Table 1.
- What is the range of high pressure in this study? Justify why ideal gas equation is valid to be used in this study.
Author response: Thank you for pointing this out. The high pressure investigated in this review is ranging from 2 MPa to 90 MPa. We assumed the ideal gas equation is applied to this pressure range.
- For GC, what’s the approximate peak time for He and Ar? Why only H2 are discussed in the section?
Author response: Thank you for pointing this out. As shown in Fig. 6(b) for GC, the approximate peak times for H2, O2 and N2 amount to 1.5 s. Because it takes 1.5 s to fill the 0.25 cc sample loop at a flow rate of 10 sccm. He in GC method was used as a carrier gas, which is not peak signal. In addition, GC equipment has only been settuped for selectively H2 gas measurement. Thus we have only provided the H2 results in GC section. We have inserted the related contents in the revised manuscript (page 9, line 262-263) as follows;
TDA-GC has been settuped for selectively H2 gas measurement. Thus we have only provided the H2 results later.
- Some of the results shown are for H2, while some are for N2. It is very confusing to follow the flow of the results and why it is presented that way.
Author response: Thank you for pointing this out. In the previous works, we have measured both uptake and diffusivity for five pure gases, such as H2, He, N2, O2 and Ar. However, we only demonstrate the results of representative pure gases (H2, N2 and O2) in the review article. We have already described the related contents in the introduction of revised manuscript.
Reviewer 2 Report
Comments and Suggestions for Authors
In this work, the author developed four different methods including the gravimetric method, volumetric method, manometric method, and gas chromatography to measure the gas uptake and diffusion of three different rubbery materials. They elaborated the principles and processes of the developed methods and compared the reliability and pros and cons of developed methods based on the results. They demonstrated that these complementary methods can be used for evaluating the leakage and sealing ability of rubbery materials and O-rings for high-pressure hydrogen fueling stations. Overall, this work would be interesting for industry and academia. A few more comments are shown below.
1. More detailed information for the sample preparation may be needed for the reproducibility of results. For example, the author measured three different samples with different shapes (spherical and cylinder): EPDM, NBR, and FKM. Are these samples house-made or brought from suppliers? If they are homemade, the author should give more information on how they are made.
2. On page 2, the author mentions the samples were degassed before exposure to test gas. Was the degassing process performed in the atmosphere or under vacuum?
3. On page 14, what are EPDM HAF40 and EPDM SRF40?
4. On page 17 line 452, “absorbed by the NBR specimen” should be “by the EPDM specimen”?
5. On page 20, the author summary the main feature of the four methods in Table 1, where it shows the GC method is a sophisticated method with complicated procedures, while in the Conclusion section (page 21 line 550), the author claims “the four developed methods are inexpensive and simple techniques….” The author may need to improve wording to avoid confusing readers.
Author Response
Reviewer 2
Comments and Suggestions for Authors
In this work, the author developed four different methods including the gravimetric method, volumetric method, manometric method, and gas chromatography to measure the gas uptake and diffusion of three different rubbery materials. They elaborated the principles and processes of the developed methods and compared the reliability and pros and cons of developed methods based on the results. They demonstrated that these complementary methods can be used for evaluating the leakage and sealing ability of rubbery materials and O-rings for high-pressure hydrogen fueling stations. Overall, this work would be interesting for industry and academia. A few more comments are shown below.
- More detailed information for the sample preparation may be needed for the reproducibility of results. For example, the author measured three different samples with different shapes (spherical and cylinder): EPDM, NBR, and FKM. Are these samples house-made or brought from suppliers? If they are homemade, the author should give more information on how they are made.
Author response: Thank you for beneficial comments. The samples used in the review were brought from suppliers. The composition and physical property of the specimens are already found in the previous literatures [25,30] of original manuscript (page 2, line 90-91).
- On page 2, the author mentions the samples were degassed before exposure to test gas. Was the degassing process performed in the atmosphere or under vacuum?
Author response: Thank you for pointing this out. The degassing process was performed by heating chamber at 343 K for more than 48 hours under the atmosphere. Thus we have inserted the related contents in the revised manuscript (page 2, line 93-94) as follows;
The heat treatment was conducted at 343 K for more than 48 hours under the atmosphere according to the CHMC 2 standard [31] to remove the outgassing from the rubber specimen.
- On page 14, what are EPDM HAF40 and EPDM SRF40?
Author response: Thank you for beneficial comments. EPDM HAF40 and EPDM SRF40 indicate the HAF (high abrasion furnace) and SRF (semi reinforcing furnace) carbon black filler of 40 phr (parts per hundred parts of rubber). The carbon black, such as HAF and SRF was normally used as a reinforcing filler in EPDM (ethylene propylene diene monomer) rubber production. Thus we have inserted the related contents in Fig. 13 caption of the revised manuscript as follows;
EPDM S20 indicates EPDM blended with silica of 20 phr (parts per hundred parts of rubber). EPDM HAF40 and EPDM SRF40 indicate EPDM blended with HAF (high abrasion furnace) and SRF (semi reinforcing furnace), respectively, carbon black filler of 40 phr.
- On page 17 line 452, “absorbed by the NBR specimen” should be “by the EPDM specimen”?
Author response: Thank you for pointing this out. In response to reviewer comments, we changed it in the revised manuscript (page 18, line 465).
- On page 20, the author summary the main feature of the four methods in Table 1, where it shows the GC method is a sophisticated method with complicated procedures, while in the Conclusion section (page 21 line 550), the author claims “the four developed methods are inexpensive and simpletechniques….” The author may need to improve wording to avoid confusing readers.
Author response: Thank you for pointing this out. To avoid the confusing readers, we have changed “the four developed methods are inexpensive and simple techniques….” into “the three developed methods are inexpensive and simple techniques, except of gas chromatography, ….” . The revised content is found in page 22, line 561-562 of revised manuscript.
Please let me know if there are still additional comments or insufficient explanation/discussion.
Thanks again for comments
With best wishes,
Jae Kap Jung

Round 2
Reviewer 1 Report
Comments and Suggestions for Authors
1. The justification/objective for this review paper is still not clear. It is suggested that the author relook into changing the word 'we' into passive voice, and rephrase the last paragraph to strengthen the justification.
2. For pressure as high as 90 MPa, there needs to be a strong justification to assume that the ideal gas equation is applied to this pressure range. Please add on the justification.
Author Response
Manuscript : polymers-2860165
Dear Editor and Reviewers,
Thank you for giving us the opportunity to submit a revised draft of the manuscript “Review of Developed Methods for Measuring Gas Uptake and Diffusivity in Polymers Enriched by Pure Gas under High Pressure” for publication in the polymers. We appreciate the time and effort that you and the reviewers dedicated to providing feedback on our manuscript and are grateful for the insightful comments on and valuable improvements to our manuscript. We have incorporated most of the comments and suggestions made by the reviewers. Those changes are represented with blue color within the manuscript for a point-by-point response to the reviewers’ comments and concerns.
Reviewer 1
- The justification/objective for this review paper is still not clear. It is suggested that the author relook into changing the word 'we' into passive voice, and rephrase the last paragraph to strengthen the justification (second comment).
Author response: Thank you for valuable comments. We changed “we” into passive voice in the abstract, introduction and conclusion. We rephrased the last paragraph of introduction as follows.
The previous research was related the diffusivity and solubility obtained from the experiments for the five gases (H2, He, N2, O2 and Ar) to the kinetic diameter and critical temperature, respectively. These gases could be appropriate candidates for the investigation of gas transport properties, because they were cheap and easily available, In this review article, the representative three types of gases (H2, N2 and O2) in the experiment and analysis were chosen. The measuring principles, measuring procedures, and representative results for three gases obtained by four methods are described. The performance and characteristics of the GM, VM, MM and GC method are reviewed. They included the investigations of various gases uptake and diffusivity obtained by methods. The comparisons among four methods, together with features of these methods, are also contained.
- What is the range of high pressure in this study? Justify why ideal gas equation is valid to be used in this study (First Comment).
- For pressure as high as 90 MPa, there needs to be a strong justification to assume that the ideal gas equation is applied to this pressure range. Please add on the justification (Second comment).
Author response: Thank you for pointing this out. The previous answer is something wrong. It is my mistake. I am very sorry for inconvenience.
The high pressure investigated in this review is ranging from 2 MPa to 90 MPa, which were the pressures for gas exposure under high pressure conditions (Fig. 1). Specimen is exposed in a high-pressure chamber (2 MPa to 90 MPa). And then after decompression to atmosphere pressure (1 atm) and loading the specimen, the emitted gas was measured by GM(Fig. 2), VM(Fig. 3), MM(Fig. 5) and GC(Fig. 6). In our situation (desorption process or emission process), the ideal gas equation using Eqs. (3), (5) and (7) was only applied to 1 atm not 90 MPa. Thus the application of ideal equation for the desorption process is valid for 1 atm.
The related content is not included in revised manuscript.
Please let me know if there are still additional comments or insufficient explanation/discussion.
Thanks again for comments
With best wishes,
Jae Kap Jung
